# The Impact of Chemotherapy on Cardiovascular Mortality across Breast Cancer Subtypes

**Toàn Minh Ngô [1,2,*], Ánh Ngọc Lê [3] and Dương Phạm Hoàng Đinh [3]**

1   Gyula Petrányi Doctoral School of Clinical Immunology and Allergology, Faculty of Medicine, University of Debrecen, H-4032 Debrecen, Hungary
2   Medical Imaging Clinic, Clinical Centre, University of Debrecen, H-4032 Debrecen, Hungary
3   Faculty of Health Sciences, University of Debrecen, H-4032 Debrecen, Hungary; lengocanh@mailbox.unideb.hu (Á.N.L.)
*   Correspondence: ngo@mailbox.unideb.hu; Tel.: +36-304-771-176

**Abstract:** Breast cancer is associated with cardiovascular mortality as an adverse effect of chemotherapy. Considering the variances across breast cancer subtypes, this study aims to investigate the cardiovascular mortality patterns in each subtype. Methods: This retrospective study used the SEER database of chemotherapy-receiving breast cancer patients (diagnosed in 2013–2020). The study population was categorized by cancer subtype, stage, patient age, and cause of death (COD). The percentage of cardiovascular CODs, odds ratio (ORs), 5-year cumulative crude probability of death, and standardized mortality ratios (SMRs) of each group were analyzed. Results: Among 23,263 nonsurviving breast cancer patients, 5.8% died from cardiovascular disease, whereas the HER2+/HR+ and HER2+/HR− subtypes exhibited the highest ORs of cardiovascular death and percentages of cardiovascular CODs, at 8.21% and 6.55%, respectively. The cardiovascular SMR increased with advancing stages and decreasing patient age. The HER2+/HR- subtype had the highest cardiovascular SMR, at 0.83 ($p < 0.05$), followed by TNBC, at 0.78 ($p < 0.05$). The 5-year cumulative probability of cardiovascular CODs also showed the highest risk in the HER2+/HR- subtype ($1.02 \pm 0.11\%$) and the TNBC subtype ($0.95 \pm 0.07\%$). Conclusion: Breast cancer patients on chemotherapy face an elevated cardiovascular mortality risk, especially with aggressive subtypes (HER2-enriched, TNBC), advanced age, or HER2+/HR+ cancer receiving long-term treatment.

**Keywords:** breast cancer; molecular subtypes; chemotherapy; cardiovascular disease





## 1. Introduction

According to the Global Cancer Statistics 2020 (GLOBOCAN), breast cancer stands out as having the highest incidence, representing 11.7% of all new cancer cases; its incidence is especially high in women, among whom it accounts for roughly a quarter of all new cancer diagnoses. Remarkably, it has the leading mortality rate (15.5%) among this population, surpassing lung cancer [1]. Over recent years, there has been a steady rise in the incidence of invasive breast cancer, especially in the early stage, attributed to increased screening and early detection efforts [2]. Fortunately, thanks to the advancements that have been made in treatment effectiveness, breast cancer mortality has improved significantly, decreasing by 43% compared to the 1990s [2].

Nonetheless, comorbidity in breast cancer presents a significant challenge, with cardiovascular disease emerging as a prominent concern [3]. Many breast cancer therapies are known to be associated with potential cardiovascular risks [4]. Moreover, recent global statistics indicate a worsening trend regarding the incidence and survival rates of cardiovascular diseases [5]. Therefore, with improvements in breast cancer treatment and enhanced survival rates, there is a growing awareness focusing on the management of potential side effects and comorbidities, with special attention directed towards addressing cardiovascular diseases.

Breast cancer can be classified into four distinct molecular subtypes based on immuno-histochemical examination of samples obtained through biopsy procedures: HER2−/HR+ (estrogen receptor (ER)/progesterone receptor (PR)-positive, human epidermal growth factor receptor 2 (HER2)-negative), HER2+/HR+ (ER/PR-positive, HER2-positive), HER2-enriched (ER/PR-negative, HER2-positive), and TNBC (triple-negative breast cancer, lacking all three receptors). Each of these subtypes is associated with unique risk profiles, prognoses, behaviors, mortality rates, and therapeutic approaches [6]. Particularly, regarding systemic chemotherapy treatment, each subtype may exhibit varying profiles of therapeutic effectiveness and side effects, and special attention should be paid to the potential impact of cardiovascular diseases.

Therefore, in this study, our primary aim was to explore differences in the rates of cardiovascular death among different patient subgroups of breast cancer patients currently undergoing chemotherapy regimens, with a particular focus on variations across the different molecular subtypes.

## 2. Materials and Methods

### 2.1. Database

This was a retrospective study using data from Surveillance, Epidemiology, and End Results (SEER), governed by the Surveillance Research Program within the National Cancer Institute's Division of Cancer Control and Population Sciences. SEER provides comprehensive statistical information on cancer cases across around the United States. For our analysis, we utilized SEER*Stat 8.4.2.

SEER Registry 17 was used; the registry covers approximately 26.5% of the U.S. population and reports over 9.2 million cases between the years 2000 and 2020. This registry encompasses patient basic demographic information including age, gender, race, ethnicity, etc.; cancer characteristics; cancer staging; treatments; vital conditions; cause of death (COD); and survival time. For the study of standardized mortality ratios, SEER Registry 17 without the geographic area of Alaska was used.

### 2.2. Study Population

The study selected female patients over 19 years old, diagnosed with breast cancer (primary tumor site ICD-O-3 codes: C500-C509) in the years 2013 to 2020 (marking the era of trastuzumab [7]), who received chemotherapy. The study exclusion criteria comprised patients with unknown ages, unknown breast cancer subtypes, or unknown cancer stages. The causes of patient deaths were categorized into breast-cancer-specific CODs (ICD-10 codes: C500-C506, C508-C509), cardiovascular CODs (including hypertensive diseases; ischemic heart disease; pulmonary heart disease and diseases of pulmonary circulation; cerebrovascular diseases; diseases of arteries, arterioles and capillaries; other and unspecified disorders of the circulatory system), and the other/remaining causes of death. Further, patients were stratified into groups based on age (20–39 years old, 40–54 years old, 55–74 years old, over 75 years old), primary cancer stage (localized, regional, distant), and breast cancer subtype (HER2+/HR−, HER2−/HR+, HER2+/HR+, TNBC).

### 2.3. Statistical Analysis

To investigate the relationship of breast cancer subtypes, different age groups, and cancer stages with cardiovascular-disease-related death, we used chi-square analysis as a univariate analysis. Additionally, using the same method, we analyzed the impact of hormone receptor positivity and HER2 positivity on cardiovascular death percentages.

We conducted a multivariate analysis, employing multinomial logistic regression, to investigate the factors influencing the cause of death, using COD as the dependent outcome. The factors considered in the analysis included breast cancer subtypes, age groups, and cancer stages.

To investigate the cardiovascular risk of each subtype relative to the general population, the standardized mortality ratios (SMRs) were calculated. The SMR is the ratio of

observed deaths by cardiovascular disease to the expected number of cardiovascular deaths in the overall United States female patient population, adjusted for age and race. The cardiovascular SMR of each age group, cancer stage, and cancer subtype was compared.

Furthermore, using the survival data in the SEER database, the crude probability of death using COD information was calculated for five years following the cancer diagnosis. This analysis investigated different causes of death acting simultaneously, specifically focusing on cardiovascular and breast-cancer-specific CODs. The probabilities of cardiovascular death and of breast cancer death were compared across all breast cancer subtypes.

All the statistical analysis was performed with SEER*Stat 8.4.2 and IBM SPSS Statistics 25.0; graphs with the standard error were generated using GraphPad Prism 9.4.1; data are presented as the mean $\pm$ standard error or the mean and 95% CI; a *p*-value of <0.05 was considered statistically significant.

## 3. Results

### 3.1. Descriptive Analysis

There were 23,263 nonsurviving breast cancer patients from 2013 to 2020, of whom most (74.04%) died from breast cancer, 5.8% from cardiovascular disease, and 20.16% from other causes of death. Among these nonsurvivors, the HER2−/HR+ cancer subtype was recorded in the highest number of breast cancer deaths (8053 deaths), followed by TNBC (5593 deaths) (Table 1). However, TNBC was the subtype with the highest percentage of breast-cancer-specific CODs (78.41%), followed by HER2+/HR− (73.35%) and HER2−/HR+ (73.14%). HER2+/HR+ had the lowest likelihood of breast-cancer-specific CODs, with a percentage of 67.49% (Figure 1a), consequently demonstrating the highest rate of non-cancer-specific causes of death. The percentage of breast cancer CODs increased with advancing cancer stages and decreased with the patient's age. However, cardiovascular COD distribution patterns differed, with the percentage of cardiovascular CODs decreasing as the cancer stage advanced and increasing with age. Remarkably, the HER2+/HR+ subtype exhibited highest percentage of cardiovascular CODs (8.21%), followed by HER2+/HR− (6.55%), while TNBC had the lowest percentage (4.26%) (Table 1).

**Table 1.** Characteristics of breast cancer patients who died from breast cancer, cardiovascular diseases, and all other causes.

| | Total | | Breast Cancer CODs | | Cardiovascular CODs | | Other CODs | |
|---|---|---|---|---|---|---|---|---|
| | *n* | % | *n* | % | *n* | % | *n* | % |
| Subtype | | | | | | | | |
| HER2+/HR− | 2075 | 100 | 1522 | 73.35 | 136 | 6.55 | 417 | 20.10 |
| HER2−/HR+ | 11,010 | 100 | 8053 | 73.14 | 660 | 5.99 | 2297 | 20.86 |
| HER2+/HR+ | 3045 | 100 | 2055 | 67.49 | 250 | 8.21 | 740 | 24.30 |
| TNBC | 7133 | 100 | 5593 | 78.41 | 304 | 4.26 | 1236 | 17.33 |
| Stage | | | | | | | | |
| Localized | 4835 | 100 | 2758 | 57.04 | 490 | 10.13 | 1587 | 32.82 |
| Regional | 10,961 | 100 | 7992 | 72.91 | 662 | 6.04 | 2307 | 21.05 |
| Distant | 7467 | 100 | 6473 | 86.69 | 198 | 2.65 | 796 | 10.66 |
| Age | | | | | | | | |
| 20–39 years | 1941 | 100 | 1670 | 86.04 | 18 | 0.93 | 253 | 13.03 |
| 40–54 years | 6180 | 100 | 5169 | 83.64 | 136 | 2.20 | 875 | 14.16 |
| 55–74 years | 12,161 | 100 | 8508 | 69.96 | 843 | 6.93 | 2810 | 23.11 |
| 75+ years | 2981 | 100 | 1876 | 62.93 | 353 | 11.84 | 752 | 25.23 |
| Total | 23,263 | 100 | 17,223 | 74.04 | 1350 | 5.80 | 4690 | 20.16 |

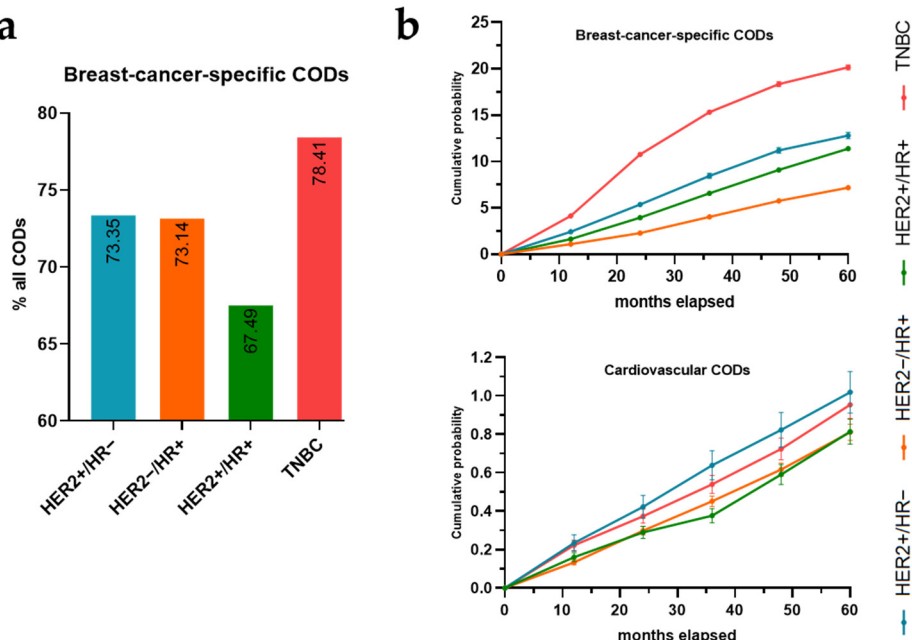

**Figure 1.** Percentages of breast-cancer-specific causes of death (CODs) (**a**); line graphs with standard error represent the cumulative probability of breast-cancer-specific CODs and cardiovascular-disease CODs over a 5-year period after the diagnosis of different breast cancer subtypes (**b**).

### 3.2. Relationship among Breast Cancer Subtype, Stage, Patient Age, and Cause of Death

Using univariate analysis, an elevated risk of breast-cancer-specific death was identified in the TNBC subtype, with an OR of 1.40 (95% CI: 1.32–1.50, $p < 0.001$), and a decreased risk was observed in the HER2+/HR+ subtype (OR: 0.69; 95% CI: 0.64–0.75, $p < 0.001$). However, regarding cardiovascular deaths, there was an increased risk in the receptor-positive groups, with an OR of 1.56 (95% CI: 1.35–1.79, $p < 0.001$) in the HER2+/HR+ group, further confirmed by ORs for cardiovascular death of 1.45 and 1.37, respectively, in the HER2-positive and HR-positive patients ($p < 0.001$) (Table 2).

**Table 2.** Odds ratios of cardiovascular disease deaths in the studied population using univariate analysis.

|  | Breast Cancer CODs | | | Cardiovascular CODs | | |
|---|---|---|---|---|---|---|
|  | OR | 95% CI | *p*-Value | OR | 95% CI | *p*-Value |
| Subtype |  |  |  |  |  |  |
| HER2+/HR− | 0.97 | 0.87–1.07 | 0.463 | 1.15 | 0.96–1.39 | 0.125 |
| HER2−/HR+ | 0.92 | 0.86–0.97 | 0.003 | 1.07 | 0.96–1.19 | 0.237 |
| HER2+/HR+ | 0.69 | 0.64–0.75 | <0.001 | 1.56 | 1.35–1.79 | <0.001 |
| TNBC | 1.40 | 1.32–1.50 | <0.001 | 0.64 | 0.56–0.73 | <0.001 |
| Receptor positivity |  |  |  |  |  |  |
| HER2-positive | 0.76 | 0.71–0.82 | <0.001 | 1.45 | 1.29–1.64 | <0.001 |
| HER2-negative | 1.31 | 1.22–1.40 | <0.001 | 0.69 | 0.61–0.78 | <0.001 |
| HR-positive | 0.75 | 0.71–0.80 | <0.001 | 1.37 | 1.23–1.55 | <0.001 |
| HR-negative | 1.33 | 1.25–1.41 | <0.001 | 0.73 | 0.65–0.82 | <0.001 |
| Stage |  |  |  |  |  |  |
| Localized | 0.36 | 0.34–0.39 | <0.001 | 2.3 | 2.05–2.59 | <0.001 |
| Regional | 0.90 | 0.84–0.95 | <0.001 | 1.08 | 0.97–1.21 | 0.146 |
| Distant | 3.06 | 2.84–3.29 | <0.001 | 0.35 | 0.30–0.40 | <0.001 |

**Table 2.** *Cont.*

| | Breast Cancer CODs | | | Cardiovascular CODs | | |
|---|---|---|---|---|---|---|
| | OR | 95% CI | *p*-Value | OR | 95% CI | *p*-Value |
| **Age** | | | | | | |
| 20–39 years | 2.29 | 2.00–2.61 | <0.001 | 0.14 | 0.09–0.22 | <0.001 |
| 40–54 years | 2.13 | 1.98–2.30 | <0.001 | 0.29 | 0.25–0.35 | <0.001 |
| 55–74 years | 0.64 | 0.60–0.68 | <0.001 | 1.56 | 1.39–1.74 | <0.001 |
| 75+ years | 0.55 | 0.50–0.59 | <0.001 | 2.6 | 2.29–2.95 | <0.001 |

A similar result was shown using multivariate analysis, with the highest OR for breast-cancer-specific death being observed in TNBC, followed by the HER2+/HR− subtype. Conversely, the OR for cardiovascular death was lowest in TNBC and highest in HER2+/HR+ cancer (OR: 1.32; 95% CI: 1.09–1.60, *p* < 0.001), followed by HER2+/HR− cancer (OR: 1.27, 95% CI: 1.01–1.61, *p* < 0.05), indicating the significant association of these two subtypes with cardiovascular fatalities (Table 3).

**Table 3.** Odds ratios from multivariate analysis of cardiovascular disease deaths among breast cancer patients' deaths with the remaining CODs as the reference.

| | Breast Cancer CODs | | | Cardiovascular CODs | | |
|---|---|---|---|---|---|---|
| | OR | 95% CI | *p*-Value | OR | 95% CI | *p*-Value |
| **Subtype** | | | | | | |
| HER2+/HR− | 0.71 | 0.62–0.81 | <0.001 | 1.27 | 1.01–1.61 | 0.041 |
| HER2−/HR+ | 0.69 | 0.64–0.75 | <0.001 | 1.17 | 1.01–1.37 | 0.042 |
| HER2+/HR+ | 0.57 | 0.51–0.63 | <0.001 | 1.32 | 1.09–1.60 | 0.005 |
| TNBC | 1 | | | 1 | | |
| **Stage** | | | | | | |
| Localized | 3.99 | 3.43–4.64 | <0.001 | 0.07 | 0.04–0.11 | <0.001 |
| Regional | 7.84 | 6.76–9.08 | <0.001 | 0.06 | 0.04–0.10 | <0.001 |
| Distant | 19.78 | 16.84–23.23 | <0.001 | 0.05 | 0.03–0.09 | <0.001 |
| **Age** | | | | | | |
| 20–39 years | 1 | | | 1 | | |
| 40–54 years | 0.93 | 0.79–1.08 | 0.317 | 2.16 | 1.30–3.60 | 0.003 |
| 55–74 years | 0.48 | 0.42–0.55 | <0.001 | 4.14 | 2.55–6.72 | <0.001 |
| 75+ years | 0.38 | 0.32–0.45 | <0.001 | 6.47 | 3.95–10.61 | <0.001 |

Investigating the impact of cancer stage on the COD, we observed a trend where advancement in the cancer stage corresponded to increased ORs for breast-cancer-specific death and decreased ORs for cardiovascular death. Conversely, the opposite trend could be seen with an increase in patient age. Specifically, the ORs for breast-cancer-specific death significantly decreased, whereas the ORs for cardiovascular death increased, suggesting a lower likelihood of death related to breast cancer but a remarkably escalated risk of cardiovascular death with advancing age. The association between patient age and cardiovascular CODs (with the relative ORs ranging from 1 to 6.47) appeared to outweigh that of the cancer stage (with the relative ORs consistently below 0.08). In contrast, a robust association was observed between breast cancer CODs and cancer stage, with the highest relative OR of 19.78 in the case of the distant stage (95% CI: 16.84–23.23, *p* < 0.001) (Table 3).

*3.3. Cumulative Crude Probability of Death by Cause of Death*

The cumulative probabilities of breast-cancer-specific CODs and cardiovascular disease CODs was examined over the 60-month study period, distinctly stratified by breast

cancer subtype. The TNBC subtype exhibited the highest probability of breast-cancer-specific CODs by a significant margin at all intervals, especially in later months, with a cumulative probability of $20.15 \pm 0.28\%$, followed by HER2+/HR− ($12.81 \pm 0.36\%$) at the 60th month. Meanwhile, with cardiovascular death as the COD, the HER2+/HR− breast cancer subtype was dominant throughout the period and peaked at $1.02 \pm 0.11\%$, followed by the TNBC subtype at $0.95 \pm 0.07\%$, whereas the HER2+/HR+ subtype consistently remained the lowest, ranging from $0.29 \pm 0.03\%$ at month 24 to $0.59 \pm 0.05\%$ at month 48 (Figure 1b).

### 3.4. Subtype-Specific Patterns of Cardiovascular Mortality across Stages and Ages

As breast cancer advances from the localized to the regional to the distant stage, a consistent decline was observed in the percentage of cardiovascular causes of death (CODs) for all breast cancer subtypes. Notably, HER2+/HR+ consistently maintained dominance in the rate of cardiovascular CODs, leading in the localized (15.77%) and regional (8.37%) stages and ranking second in the distant stage (3.11%), following HER2+/HR− (3.29%). In contrast, TNBC consistently exhibited the lowest percentages of cardiovascular CODs relative to overall CODs across all three stages, with the percentage of cardiovascular CODs decreasing from 8.29% in the localized stage to 3.25% in the regional stage and further dropping to 2.23% in the distant stage (Figure 2a).

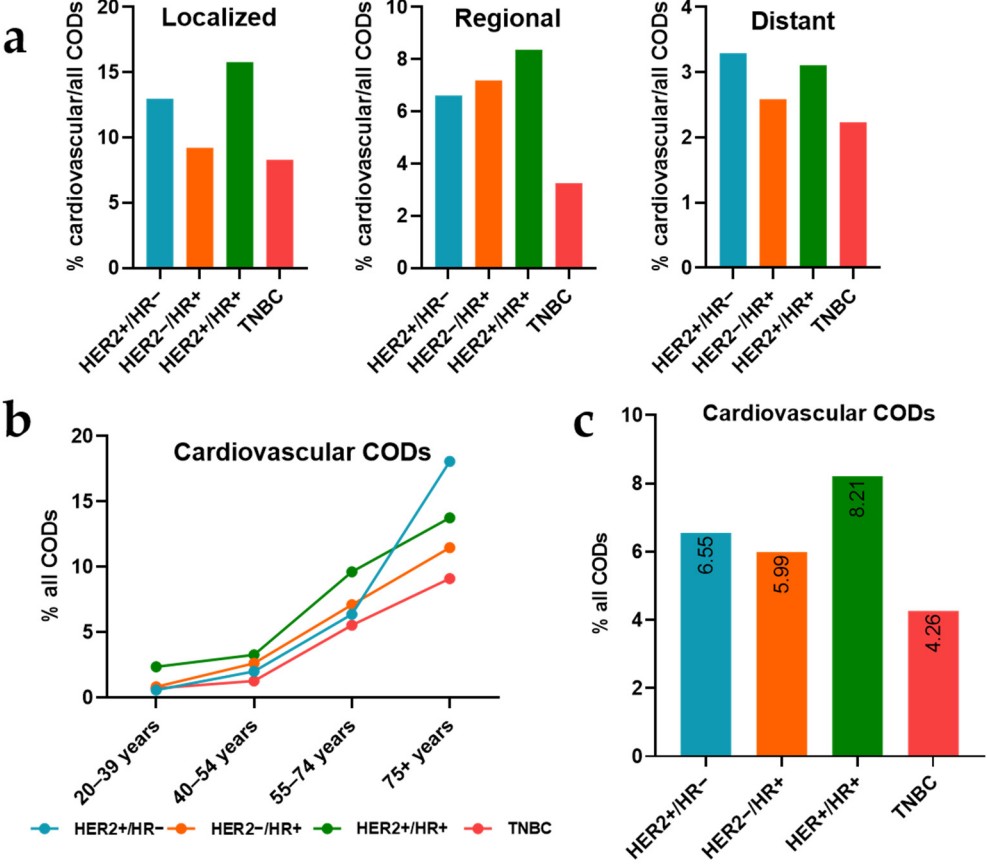

**Figure 2.** Percentage (%) of cardiovascular CODs among all CODs of patients with different breast cancer subtypes in each stage (**a**), % of cardiovascular CODs among all CODs of patients with different breast cancer subtypes in each age group (**b**), % of cardiovascular CODs among all CODs of patients with different breast cancer subtypes in all the age groups and stages (**c**).

The trend of older patients consistently exhibiting increased mortality from cardiovascular CODs was evident across all breast cancer subtypes. Specifically, HER2+/HR+ maintained its place as the subtype with the highest proportion of cardiovascular CODs

across most of the age groups, particularly at 20–29 (2.35%), 40–54 (3.27%), and 55–74 years of age (9.63%). There was a significant increase in the percentage of cardiovascular CODs is observed in patients with the HER+/HR− subtype at ages 55–74 years, reaching 6.38%, and peaking at 18.07% for the age group of 75+ years (Figure 2b).

Overall, among breast cancer subtypes, HER2+/HR+ and HER2+/HR− demonstrated the highest percentages of cardiovascular causes of death (CODs), at 8.21% and 6.55%, respectively (Figure 2c).

### 3.5. Subtype-Specific Cardiovascular SMRs across Stages and Age Groups

Cardiovascular standardized mortality ratios based on stages, age groups, and subtypes of cancer varied differently. The SMR values increased along with the development of cancer from stage to stage, from 0.58 (95% CI: 0.52–0.65, $p < 0.05$) in the localized stage to 0.79 (95% CI: 0.73–0.87, $p < 0.05$) in the regional stage, and further to 1.56 (95% CI: 1.32–1.84, $p < 0.05$) in the distant stage. Patients with localized cancer of the TNBC subtype had the highest SMR for cardiovascular mortality, with a value of 0.64 (95% CI: 0.53–0.78, $p < 0.05$), followed by the HER2+/HR− subtype, with 0.63 (95% CI: 0.45–0.86, $p < 0.05$). A similar pattern was observed in the distant stage, where the TNBC subtype was dominant, with an SMR of 2.60 (95% CI: 1.77–3.69, $p < 0.05$), followed by the HER2+/HR− subtype, with an SMR of 2.06 (95% CI: 1.27–3.14, $p < 0.05$). The SMR of the HER2−/HR+ subtype remained the lowest throughout the regional and distant stages (Figure 3a).

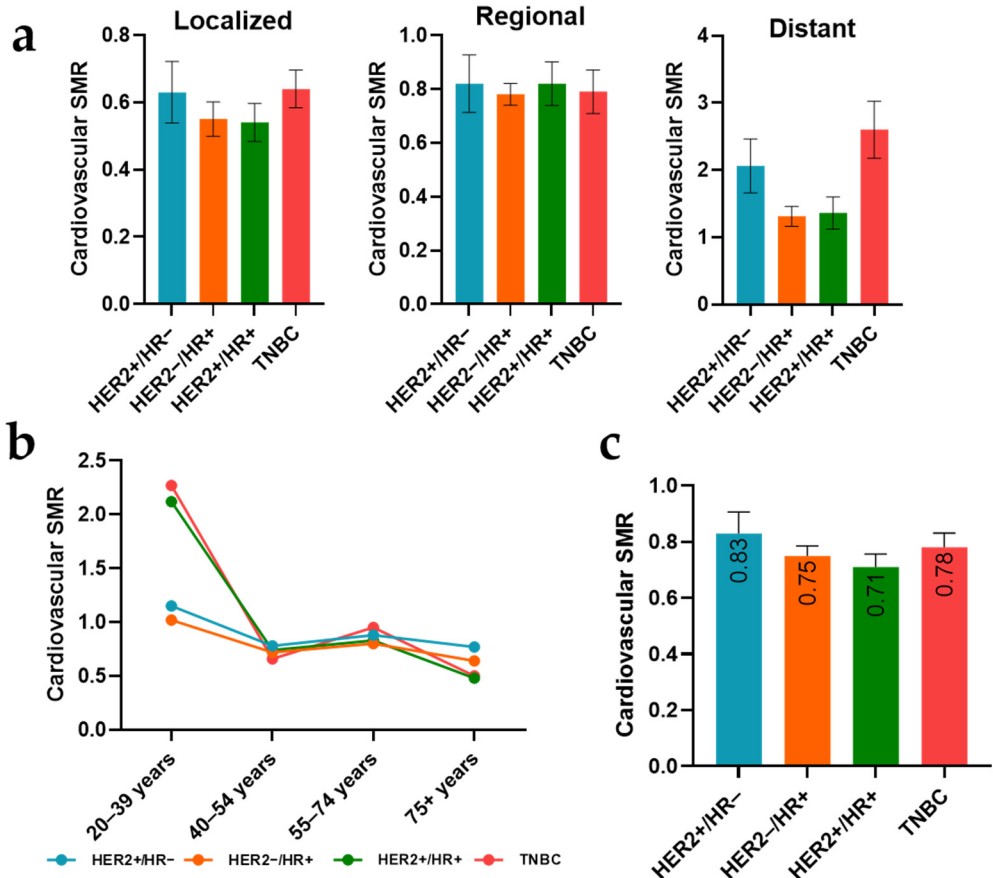

**Figure 3.** Standardized mortality ratios (SMRs) of different breast cancer subtypes in each stage (**a**), SMRs of different breast cancer subtypes in each age group (**b**), SMRs of different breast cancer subtypes in all the age groups and stages (**c**). Bar graphs represent mean and standard error.

Unlike the percentage of cardiovascular CODs among all CODs across different breast cancer subtypes in each age group, there was a considerable decline in the cardiovascular SMR among subtypes as age increased. In the age range of 55–74 years, which constituted

the largest population in our study, patients with the TNBC breast cancer subtype exhibited the highest SMR, at 0.95 (95% CI: 0.80–1.11), followed by the HER2+/HR− subtype, with an SMR of 0.88 (95% CI: 0.67–1.14). In the group of elderly patients (75+ years), who suffer the highest risk of cardiovascular events, the HER2+/HR− subtype had the highest SMR, at 0.77 (95% CI: 0.55–1.04), whereas the HER2+/HR+ subtype had the lowest SMR, at 0.48 (95% CI: 0.35–0.63, $p < 0.05$) (Figure 3b).

Considering both age groups and stage factors among breast cancer subtypes, patients with the HER2+/HR− subtype had the highest SMR of death from heart disease, at a value of 0.83 (95% CI: 0.68–1.00, $p < 0.05$), followed by TNBC, at 0.78 (95% CI: 0.68–0.89, $p < 0.05$), and then by HER2−/HR+ and HER2+/HR+, at 0.75 (95% CI: 0.68–0.82, $p < 0.05$) and 0.71 (95% CI: 0.62–0.82, $p < 0.05$), respectively (Figure 3c).

## 4. Discussion

Distinct breast cancer subtypes necessitate specific systemic therapies [8]. In addition to the conventional chemotherapy agents, such as alkylating drugs, antimetabolites, plant alkaloid, and antineoplastic antibiotics, hormone-receptor-positive tumors benefit from endocrine treatments such as selective estrogen receptor modulators, e.g., tamoxifen or aromatase inhibitors, whereas HER2-positive cancers require targeted therapy with agents such as monoclonal antibodies against HER2, e.g., trastuzumab. Despite their efficacy, these treatments pose significant cardiovascular risks. Specifically, tamoxifen elevates the thromboembolic risk, while aromatase inhibitors decrease estrogen levels and consequently diminish the estrogenic protective effect against cardiovascular events. 5-Fluorouracil increases the risk of ischemic heart disease, arrhythmia, and heart failure. Taxane drugs induce cardiotoxicity by making structural changes to myocytes and causing conduction problems. Doxorubicin, cisplatin, and cyclophosphamide are known for their cytotoxicity, eventually leading to myocyte death. Trastuzumab is widely known for its adverse effect on heart failure [4,9]. Therefore, precision patient selection and dedicated monitoring of therapies are needed, particularly considering the diverse cardiovascular risks associated with different breast cancer subtypes.

The HER2−/HR+ subtype, which has emerged as the most frequently diagnosed [10–14], accounted for the highest number of deaths in our study. However, the most lethal subtypes are consistently identified as TNBC and HER2, as shown in numerous studies [11–14]. In our study, data on both the percentage of breast-cancer-specific CODs and the cumulative probability of death indicated that TNBC was the deadliest, followed by the HER2 subtype. Furthermore, there was notably a strong connection between breast-cancer-specific death and the TNBC subtype, particularly in the advanced stage and in the youngest age group. This correlation is attributed to the fact that TNBC, the most lethal subtype, exhibits the highest incidence among all subtypes in the youngest age cohort [15].

Many studies have found a strong association between breast cancer and cardiovascular risk, particularly in the context of chemotherapy [16,17]. Within our study, we observed subtype heterogeneity concerning cardiovascular deaths. Notably, the HER2+/HR+ subtype exhibited the highest ratio of cardiovascular causes of death to all causes of death, with HER2 and HR positivity correlating with an increased risk of cardiovascular death, further accentuated with age [18] and declining with successive stages. Furthermore, the HER2+/HR+ subtype had the highest comorbidity burden, and these comorbidities significantly influenced overall survival in this subtype [12,19].

Nevertheless, owing to the longer survival and less aggressive nature of the HER2+/HR+ subtype, the risk of cardiovascular events can be influenced by the increasing age of patients. To address this, we applied SMRs to examine cardiovascular mortality among subtypes, adjusting for the general population. There was a decline in SMRs across all subtypes as age advanced, with the most distinct disparities in SMRs observed among subtypes in the distant stage. In comparison to the proportion of cardiovascular deaths among all causes, TNBC exhibited a notable increase in cardiovascular SMRs compared to the other subtypes, while the luminal HER2+/HR+ subtype had the lowest SMRs. This disparity may be due

to not only the more indolent nature of HER2+/HR+ but also to the difference in the age distributions of breast cancer subtypes. Luminal subtypes are more prevalent in older patients, who have a higher risk of cardiovascular events in the general population, while the TNBC subtype's incidence decreases with increased age [15,20].

Differences can be observed across various stages when comparing cardiovascular SMRs and the percentage of cardiovascular CODs. Specifically, cancer in the early stages demonstrates a higher cardiovascular percentage among CODs, attributed to therapy side effects, primarily caused by radiotherapy [2]. Moreover, early-stage breast cancer exhibits fewer deaths specific to breast cancer, resulting in a higher percentage of deaths from comorbidities, among which cardiovascular deaths predominate [21]. Conversely, the occurrence of the highest SMRs in the distant stage reflects a more intensive chemotherapy regimen and elevated cardiovascular risk compared to the general population. Additionally, both the 5-year crude death rate from cardiovascular CODs and the cardiovascular SMRs showed that the more aggressive subtypes entailed a higher cardiovascular risk. Notably, the HER2 subtype, despite exhibiting a better survival rate than TNBC, poses a higher cardiovascular risk. This may be attributed to the significant cardiovascular adverse effects of trastuzumab, in addition to conventional chemotherapy, as part of the treatment for the HER2 subtype.

Fortunately, many clinical guidelines for managing cardiovascular risk for breast cancer patients have been proposed [22–24], including the recent cardio-oncology guidelines from the International Cardio-Oncology Society (ICOS) and the European Society of Cardiology (ESC) [25]. It is recommended that, prior to initiating therapy, patients should undergo a comprehensive assessment of medical history and a physical examination, complemented by cardiac biomarker analysis (cardiac troponin, natriuretic peptides) and imaging modalities (strain echocardiography, cardiac MRI, radionuclide angiography) to establish baseline risk and to enable future stratification of the risk of cardiovascular adverse effects during and after therapy. These evaluations enable the cardio-oncology team to provide breast cancer patients with timely and optimal prevention methods utilizing various cardioprotective agents, such as beta blockers, statins, aldosterone antagonists, ACE inhibitors, dexrazoxane, etc. [22–25]. Moreover, effective evaluation and timely intervention offer the potential to improve cardiac function, even in challenging cases of chemotherapy-induced cardiotoxicity, particularly with anthracycline, which induces cell death and is commonly regarded as irreversible [26,27]. Consequently, the latest clinical trials have demonstrated positive results in managing cardiovascular adverse effects and minimizing unnecessary cancer-therapy interruptions in breast cancer patients [22], leading to a recent decline in heart disease mortality trends among this high-risk population [17]. Nonetheless, despite rapid advancements in the field of cardio-oncology, there is an ongoing need for improved risk evaluation and innovative cardioprotective strategies to reduce the morbidity and the mortality associated with cardiovascular disease in patients with breast cancer.

This study has several limitations. Firstly, we did not analyze the specific effects of individual breast cancer medications on cardiovascular mortality, which is particularly relevant when different drugs have distinct cardiotoxicity profiles [28]. Our study lacked the capability to analyze different cardiovascular CODs, especially as breast cancer patients have variable incidence and mortality patterns—specifically, the highest ischemic heart disease death rate among breast cancer patients but the highest heart failure and hypertension mortality risks when compared to the control group [29]. Secondly, the study was admittedly unable to fully capture the rapid advancements in cardiovascular management in oncology care over the study period, which may have significantly influenced patient outcomes. Another limitation of this study was the omission of consideration for the potential effects of radiotherapy, which can be an important aspect given its relevance to cancer treatment and its potential influence on cardiovascular health [17]. Future studies should categorize specific cardiovascular causes of death, examine the individual effects of different cancer drugs, and consider the impact of radiation therapy. This approach

would enhance our understanding of the relationship between breast cancer treatment and cardiovascular outcomes.

**Author Contributions:** T.M.N. contributed to the main conceptualization, methodology, result interpretation, review, and editing. Á.N.L. and D.P.H.Đ. contributed to the data acquisition, data analysis, and original draft preparation. All authors have read and agreed to the published version of the manuscript.

**Funding:** This research received no external funding.

**Institutional Review Board Statement:** The requirements for ethical review and approval were waived for this study because the data in this retrospective study are from a public database, the National Cancer Institute-funded Surveillance, Epidemiology, and End Results (SEER) database (https://seer.cancer.gov/) accessed on 2 November 2023.

**Informed Consent Statement:** As the data in this retrospective study are from a public database, formal consent was not required.

**Data Availability Statement:** Publicly available datasets were analyzed in this study. These data can be found here: https://seer.cancer.gov/ (accessed on 2 November 2023).

**Acknowledgments:** We would like to express our appreciation to the University of Debrecen for their generous support in providing a publication discount as a member of the International Organization of Publishers and Authors (IOPA).

**Conflicts of Interest:** The authors declare no conflicts of interest.

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
