# Peer review of "The Impact of Chemotherapy on Cardiovascular Mortality across Breast Cancer Subtypes"

_curroncol, doi:10.3390/curroncol31020047_

Round 1
Reviewer 1 Report
Comments and Suggestions for Authors
Dear authors,
I have reviewed your article " The impact of chemotherapy on cardiovascular mortality across breast cancer subtypes".
Please see the comments below for the details of the peer review.
I look forward to better submissions in the future.
I have several comments.
1. Lines 116-117: “HER2+/HR+ had the lowest likelihood of breast cancer-specific COD, with a percentage of 67.49% (Figure 1a), but showed the highest rate of the non-cancer specific cause of death.” The lowest rate of breast cancer-specific COD will inevitably result in the highest rate of non-cancer-specific causes of death.
2. Lines 155-162: “Investigating the impact of cancer stage on the COD, we observed a trend where advancing in the cancer stage corresponded to increased ORs for breast-cancer specific death and decrease ORs for cardiovascular death. Conversely, an opposite trend can be seen with an increase in patient age. Specifically, the ORs for breast cancer specific death significantly decreased whereas ORs for cardiovascular death increased, suggesting a lower likelihood of deaths related to breast cancer but an escalating risk of cardiovascular death with advancing age, even though the association between patients age and cardiovascular COD was only subtle (with the relative ORs consistently below 0.5) (Table 3).” The above text is the same as in Lines 144-151.
3. Please modify the ORs listed in Table 3 for each age group to be based on ‘20-39 years’.
4. Please calculate the OR of cardiovascular risk with each anticancer drug.
5. Please describe the percentage of venous thromboembolism and heart failure among cardiovascular COD.
Date of this review
10th Jan. 2024
Author Response
We really appreciate your time and effort to review this manuscript.
Please see the attachment.

Reviewer 2 Report
Comments and Suggestions for Authors
The authors have done a good job regarding the relevant issue of CV mortality in BC patients. The article is generally well written and I have only a couple of minor issues:
1) There should be a limitations section in the discussion
2) The authors discussion of the potential CV effects of hormone therapy or trastuzumab are laid out as if nothing could be done to treat/prevent or attenuate its effects, nor is the relevant issue of Cardio-Oncology mentioned even once. Breast cancer was one of the main drivers of the emergence of this field and a great deal has been done since to reduce CV mortality in these patients. The authors should acknowledge that a 2013-2020 timeframe likely represents different patterns regarding cardiotoxicity prevention/treatment, and thus this is a relevant limitation. Furthermore, they must emphasize the importance of Cardio-Oncology in addressing this relevant issue a lot more, not only mentioning recent papers in the field, but also arguably current guidelines on the topic (ICOS/ESC). The authors may, lastly, argue that despite this a lot of further research to improve outcomes is necessary, especially in younger patients
Author Response
We are really grateful for your time and effort to review our manuscript.
Please see the attachment.

Round 2
Reviewer 1 Report
Comments and Suggestions for Authors
Dear authors,
I have reviewed your revised article.
Please see the following comments.
I look forward to better submissions in the future.
I have several comments
1. Lines 152-154 and Table 3: Please modify the ORs for each stage to be based on ‘Localized’.
2. Lines 296-298: “These evaluations enable the cardio-oncology team to provide breast cancer patients with timely and optimal prevention methods utilizing various cardio protective agents, such as beta blockers, statins, aldosterone antagonists, ACE inhibitors, dexrazoxane, etc. [22-25].” Proper evaluation not only prevents cardiovascular adverse events, but may also lead to treatment (PMID: 37180802).
Date of this review
19th Jan. 2024
Author Response
Thank you for your prompt review. Please see the attachment.

Round 3
Reviewer 1 Report
Comments and Suggestions for Authors
Thank you for your careful answers to my questions.
I hope this manuscript will be published soon.